# DT+GNN: A Fully Explainable Graph Neural Network using Decision Trees

## Abstract

We propose a new Decision Tree Graph Neural Network (DT+GNN) architecture for Graph Neural Network (GNN) explanation. Existing post-hoc explanation methods highlight important inputs, but fail to reveal how a GNN uses these inputs. In contrast DT+GNN is fully explainable: Humans can inspect and understand the decision making of DT+GNN at every step. DT+GNN internally uses a novel GNN layer that is restricted to categorical state spaces for nodes and messages. After training with gradient descent, we can distill these layers into decision trees. These trees are further pruned using our newly proposed method to ensure they are small and easy to interpret. DT+GNN can also compute node-level importance scores like the existing explanation methods. We demonstrate on real-world GNN benchmarks that DT+GNN has competitive classification accuracy and computes competitive explanations. Furthermore, we leverage DT+GNN's full explainability to inspect the decision processes in synthetic and real-world datasets with surprising results. We make these inspections accessible through an interactive web tool.

## 1 Introduction

Graph Neural Networks (GNNs) have been successful in applying machine learning techniques to many graph-based domains [5; 20; 56; 28]. However, current GNNs are black-box models, and their lack of human interpretability limits their use in many application areas. This motivated the adoption of existing deep learning explanation methods [4; 25; 41] to GNNs, and creating new GNN-specific methods [59; 51]. Such methods allow us to understand *what* parts of the input were important for making a prediction. However, it usually remains difficult to impossible for users to understand *why* these parts were important.

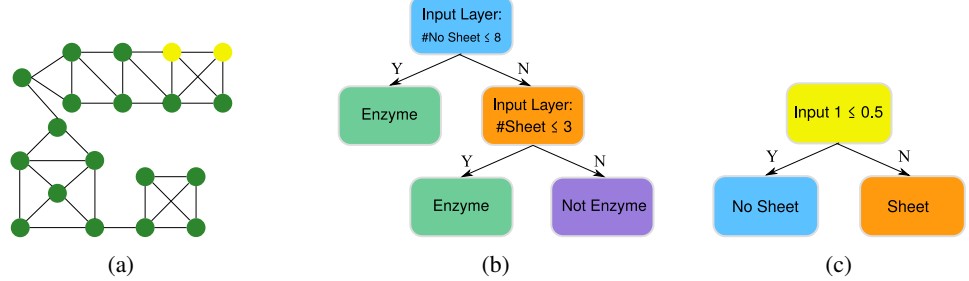

Figure 1: (a) An example graph in the PROTEINS dataset. In this dataset nodes are secondary structural elements of amino acids. Each node has one of three types: helix (input 0), sheet (input 1), or turn (input 2). Figure (a) shows an example graph consisting of mostly helixes (green) and two sheets (yellow). The final layer (b) learns to decide whether a protein is an enzyme based on how many sheet respectively "no sheet" nodes the graph has in the input layer. Whether nodes are helixes or turns does not seem to matter, and consequently the input layer (c) learns to only distinguish between sheet and no sheet. Both trees (b,c) together then imply: *A protein is an enzyme if it has at most* 8 *nodes that are not sheets, or if it has at most* 3 *sheets.* This DT+GNN explanation is consistent with previous human explanations [16].

As a motivating example, consider the PROTEINS example in Figure 1a. Previous methods classify the example as an enzyme, and mark the two yellow nodes in Figure 1a as important. But why are the two yellow nodes important? Is it because they are connected? Do we need exactly two yellow nodes? Or could there be three as well? Existing explanation methods that compute importance scores do *not* answer such questions, and as such humans have to try to figure out the decision rules by carefully looking at dozens of examples.

We overcome this limitation with a new fully explainable GNN architecture called Decision Tree GNN (DT+GNN). Figures 1b and 1c show how DT+GNN describes the decision process (including all thresholds) on the PROTEINS dataset. This explanation goes beyond understanding individual examples. Instead DT+GNN explains how the GNN solves the whole dataset.

For PROTEINS, DT+GNN can explain the decision with two simple decision trees. But DT+GNN can also fully explain datasets such as BA-2Motifs [35] or Tree-Cycle [59]. These advanced datasets combine multi-layer reasoning typical for GNNs, including detecting neighborhood patterns and degree counting.

In summary, our contributions are as follows:

- We propose a new differentiable Diff-DT+GNN layer. While traditional GNNs are based on synchronous message passing [32], our new layer is inspired on a simplified distributed computing model known as the stone age model [15]. In this model, nodes use a small categorical space for states and messages. We argue that the stone age model is more suitable for interpretation while retaining a high theoretical expressiveness.

- We distill all Multi-Layer Perceptrons that Diff-DT+GNN uses internally to Decision Trees. Our new model is called DT+GNN (decision tree graph neural network); it consists of a series of decision tree layers. Our new model is fully explainable, because one can just follow the decision tree(s) to understand a decision.

- We propose a way to collectively prune the decision trees in DT+GNN without compromising accuracy. This leads to smaller trees, which further increases explainability. We can further use these trees to compute node-level importance scores similar to current explanation methods.

- We test our proposed architecture on established GNN explanation benchmarks and real-world graph datasets. We show that our models are competitive in classification accuracy with traditional GNNs and competitive in explanation accuracy with existing explanation methods. We further validate that the proposed pruning methods considerably reduce tree sizes. Also, we demonstrate DT+GNN's full explainability to discover problems in existing explanation benchmarks and to find interesting insights into real-world datasets.

- We provide a user interface for DT+GNN.[1] This tool allows for the interactive exploration of the DT+GNN decision process on the datasets examined in this paper. We provide a manual for the interface in Appendix A.

## 2    RELATED WORK

### 2.1    EXPLANATION METHODS FOR GNNS

In recent years, several methods for providing GNN explanations were proposed. These methods highlight which parts of the input are important in a GNN decision, usually by assigning importance scores to nodes and edges or by finding similar predictions. These scores and examples can assist humans to find patterns that might reveal the GNN decision process. The existing explanation methods can be roughly grouped into the following six groups:

**Gradient based.** Baldassarre & Azizpour [4] and Pope et al. [41] show that it is possible to adopt gradient-based methods that we know from computer vision, for example Grad-CAM[48], to graphs. Gradients can be computed on node features and edges [47].
**Mutual-Information based.** Ying et al. [59] measure the importance of edges and node features. Edges are masked with continuous values. Instead of gradients, the authors use mutual information between the inputs and the prediction to quantify the importance. Luo et al. [35] follow a similar idea

---

[1]https://interpretable-gnn.netlify.app/

but emphasize finding structures that explain multiple instances at the same time.

**Subgraph based.** Yuan et al. [61] consider each subgraph as possible explanations. To score a subgraph, they use Shapley values [50] and monte carlo tree search for guiding the search. Duval & Malliaros [14] build subgraphs by masking nodes and edges in the graph. They run their subgraph through the trained GNN and try to explain the differences to the entire graph with simple interpretable models and Shapley values. Zhang et al. [63] infer subgraphs called prototypes that each represent one particular class. Graphs are classified and explained through their similarity to the prototypes.

**Example based.** Huang et al. [25] proposes a graph version of the LIME [43] algorithm. A prediction is explained through a linear decision boundary built by close-by examples. Vu & Thai [51] aim to capture the dependencies in node predictions and express them in probabilistic graphical models. Faber et al. [17] explain a node by giving examples of similar nodes with the same and different labels. Dai & Wang [11] create a $k$-nearest neighbor model and measure similarity with GNNs.

**Counterfactual.** Counterfactual approaches measure the importance of nodes or edges by how much removing them changes the classifier prediction. They are the extension of the occlusion idea [62] to graph neural networks. Lucic et al. [33] for example identifies few edge deletions that change model predictions. Bajaj et al. [3] propose a hybrid with an example-based explanation. They compute decision boundaries over multiple instances to find optimized counterfactual explanations.

**Simple GNNs.** Another interesting line of research is simplified GNN architectures [8; 9; 24]. The main goal of this research is to show that simple architectures can perform competitively with traditional, complex ones. As a side effect, the simplicity of these architectures also makes them slightly more understandable.

Complimentary to the development of explanations methods is the research on how we can best evaluate them. Sanchez-Lengeling et al. [45] and Yuan et al. [60] discuss desirable properties a good explanation method should have. For example, an explanation method should be faithful to the model, which means that an explanation method should reflect the model's performance and behavior. Agarwal et al. [1] provide a theoretical framework to define how strong explanation methods adhere to these properties. They also derive bounds for several explanation methods. Faber et al. [18] and Himmelhuber et al. [22] discuss deficiencies in the existing benchmarks used for empirical evaluation. We will pick up on these deficiencies and use DT+GNN to confirm such flaws in Sections 4.3 and D.

Note that all these methods do not provide *full* explainability. They can show us the important nodes and edges a GNN uses to predict. But (in contrast to DT+GNN) they cannot give insights into the GNN's decision-making process itself.

## 2.2 DECISION TREES FOR NEURAL NETWORKS

Decision trees are powerful machine learning methods, which in some tasks rival the performance of neural networks [21]. The cherry on top is their inherent interpretability. Therefore, researchers have tried to distill neural networks into trees to make them explainable [7; 10; 12; 31]. More recently, Schaaf et al. [46] have shown that encouraging sparsity and orthogonality in neural network weight matrices allows for model distillation into smaller trees with higher final accuracy. Wu et al. [53] follow a similar idea for time series data: they regularize the training process for recurrent neural networks to penalize weights that cannot be modeled easily by complex decision trees. Yang et al. [57] aim to directly learn neural trees. Their neural layers learn how to split the data and put it into bins. Stacking these layers creates trees. Kontschieder et al. [30] learn neural decision forests by making the routing in nodes probabilistic and learning these probabilities and the leaf predictions.

Our DT+GNN follows the same underlying idea: we want to structure a GNN in a way that allows for model distillation into decision trees to leverage their interpretability. However, the graph setting brings additional challenges. We not only have feature matrices for each node, but we also need to allow the trees to reason about the state of neighboring nodes one or more hops away.

A very recent work from Aytekin [2] shows that we can transform any neural network into decision trees. However, this approach creates a tree with potentially exponentially many leaves and it does not create understandable messages. This prohibits explainability further since we cannot undestand the message passing steps. DT+GNN addresses both problems by allowing losses in accuracy for conciseness and using categorical states. The latter decomposes the decision process by layers and creates understandable messages.

## 3 THE DT+GNN MODEL

### 3.1 CREATING A TREE-BASED GNN MODEL

Our idea to create a fully-explainable model is to build a GNN which is composed of decision trees. We start our architecture from a GIN model Xu et al. [55] with $\epsilon = 0$. The GIN aggregation rule applies a parametrizable function $f_\theta$ on the node state and a sum over the node's neighbors' states:

$$h^{l+1}(v) = f_\theta(h^l(x), \sum_{w \in Nb(v)} h^l(w))$$

Additionally, our GIN model has an encoder layer for its input features and a decoder layer to map the final embeddings $h^L(v)$ to class probabilities using skip connections. For node classification, the decoder has access to the embeddings of each layer. For graph classification, the decoder has access to the pooled embeddings of each layer.

However, the GIN model is not interpretable since the intermediate states $h^l(v)$ are continuous embeddings. This makes it hard to understand what information is passed around between the nodes in each communication step. Loukas [32] shows that GNNs such as GIN operate in a similar manner to synchronous message passing algorithms from distributed computing. Often, these algorithms have a limit on the message size of $b = O(\log n)$ bits (where $n$ is the number of nodes) but can perform arbitrary local computation [40]. In contrast to this, the stone age distributed computing model [15] assumes that each node uses a finite state machine to update its state and send its updated state as a message to all neighbors. The receiving node can count the number of neighbors in each state. A stone age node cannot even count arbitrarily, it can only count up to a predefined number, in the spirit of "one, two, many". Neighborhood counts above a threshold are indistinguishable from each other. Interestingly enough, such a simplified model can still solve many distributed computing problems [15].

We build a fully differentiable Diff-DT+GNN layer following this stone-age model. To do this, we extend the GIN update rule with a Gumbel-Softmax [27; 36] to produce categorical states. Formally:

$$h^{l+1}(v) = Gumbel(f_\theta(h^l(x), \sum_{w \in Nb(v)} h^l(w)))$$

Furthermore, we also apply a Gumbel-Softmax to the result of the encoder layer. Therefore, hidden states become one-hot vectors and as a consequence the summation in the GIN update rule is now counting the number of neighbors in each state — just like in stone age. Unlike stone age, we did not find that limiting the counting gave better results or better interpretability.

From a theoretical perspective, the categorical state space of Diff-DT+GNN layers does not reduce expressiveness. If we have a GNN layer with continuous embeddings that encode $O(\log n)$ bits of information, we can construct a Diff-DT+GNN layer using $O(n)$ bits that can represent the same information. Practically, a state space with thousands of states is not tractable for a human and not interpretable. Therefore, we will constrain the Diff-DT+GNN to few ($O(n)$) categorical states. In theory we look at an exponential losss in expressiveness. However, we noticed that in practice we incur hardly a loss for many datasets (c.f., Table 1a).

Next, we replace the update rule in each Diff-DT+GNN layer by a decision tree. We find these trees through model distillation: we pass all of the training graphs through the Diff-DT+GNN model and record all node states $h^l(v)$. The tree for layer $l$ learns to predict $h^{l+1}(v)$ from $h^l(v)$ and $\sum_{w \in Nb(v)} h^l(w)$ over all nodes. Since node states are categorical in Diff-DT+GNN this distillation is a classification problem. Formally our DT+GNN layer computes:

$$h^{l+1}(v) = TREE^l(h^l(x), \sum_{w \in Nb(v)} h^l(w))$$

There is one caveat with using decision trees versus MLPs. Unlike MLPs, decision trees cannot easily compare two features, for example, to find out if one feature is larger than the other. To produce small trees we help the decision trees with comparisons: We include pairwise delta features $\Delta$, binary features that compare every pair of states. Let $c^l(v) = \sum_{w \in Nb(v)} h^l(w)$ be vector containing the numbers of neighbors of each state in layer $l$ and let S be the set of states. Then we compute the delta

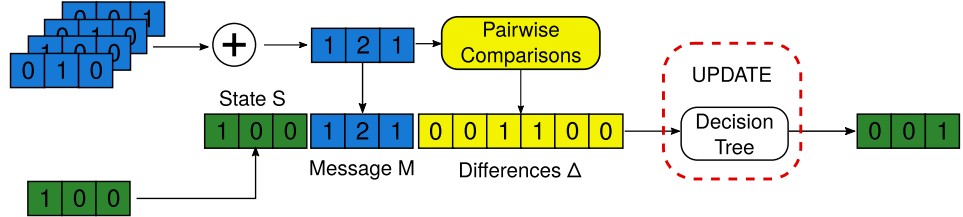

Figure 2: A DT+GNN layer. DT+GNN updates the state of a node based on its previous categorical state (here 0), the number of neighbors per state (1 in state 0, 2 in state 1, 1 in state 2), and binary $>$ comparisons between states (only state 1 outnumbers other states, therefore the third and fourth deltas are 1). A decision tree receiving this information computes the followup categorical state (here 2).

features as bits $b_{ij}$ between pairs of states if the number of neighbors in state $i$ outnumber the number of neighbors in state $j$. We compute these $Delta$ features for tree training and during inference. Our final update rule for a DT+GNN layer as shown in Figure 2 is:

$$\Delta(c^l) = \underset{i \in S, j \neq i \in S}{\|} \mathbb{1}_{c_i > c_j}$$
$$h^{l+1}(v) = TREE^l(h^l(x), c^l(v), \Delta(c^l(v)))$$

After training, we inspect the features that the tree uses. If the tree uses one of the first three features $x_0, x_1$, or $x_2$, the tree considered the previous state of the node. We model this decision node as in Figure 3a. If the tree uses one of the next three features $x_3, x_4$, or $x_5$, the decision tree split is based on the neighborhood count of one particular state (shown in Figure 3b). The threshold $y$ is found by the decision tree during training. If the feature is one of the remaining six, the decision split is using one of the pairwise comparison features, which we can model as in Figure 3c.

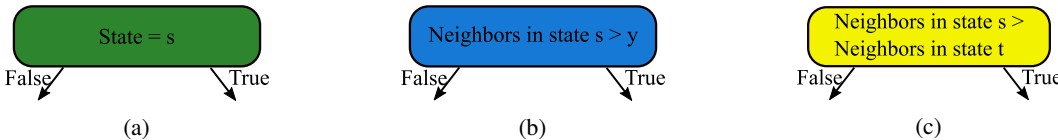

Figure 3: The different branches possible in a DT+GNN layer. We can branch on (a) the state a node is in, (b) if the node has a certain number of neighbors in a certain state, or (c) if the node has more neighbors in one state than another state.

We distill the encoder and decoder in the same way. Overall, DT+GNN still follows the stone age model and is also fully interpretable. When we see a layer use a state from an intermediate layer we can recover the semantic meaning of this state. We can recursively look up the decision tree for that intermediate layer to understand when DT+GNN assigns nodes to this state. Potentially, to fully understand the decision process we need to recurse through multiple layers. However, since every layer is a decision tree we always find an interpretable model at each layer.

## 3.2 POSTPROCESSING THE DT+GNN

**Pruning trees.** Decision trees like MLPs can theoretically be universal function approximators if they are sufficiently deep [44]. But again we prefer interpretable small and shallow decision trees. Shallow trees are more akin to the finite state machine used in the stone age distributed computing model. One easy solution that we employ during training is to restrict the number of decision leaves per tree. However, we found that this is not sufficient since the decision trees contain many nodes that are artifacts caused by overfitting the training set.

We prune these nodes based on the reduced error pruning algorithm [42]. First, we define a set of data points used for pruning (pruning set). Then, we assess every decision node, starting with the one that acts upon the fewest data points. If replacing the decision node with a leaf does not lead to an accuracy drop over the entire DT+GNN on the pruning set, we replace it. We keep iterating through all decision nodes until no more changes are made.

Choosing the pruning set is not trivial. If we use the training set for pruning, all of the overfitted edge cases with few samples are used and not pruned. On the other hand, the much smaller validation set might not cover all of the decision paths and cause overpruning. Therefore, we propose to prune on both training and validation sets with a different pruning criterion: A node can be pruned if replacing it with a leaf does not reduce the validation set accuracy (as in reduced error pruning) and does not reduce the training set accuracy below the validation set accuracy. Not allowing validation accuracy to drop ensures that we do not overprune. But since we allow a drop in training accuracy for the modified tree, we also remove decision nodes that result from overfitting.

Empirically, we found that we can prune substantially more decision nodes when we allow for a slight deterioration in the validation accuracy. We follow the same approach as before and always prune the decision node that leads to the smallest deterioration. We repeat this until we are satisfied with the tradeoff between the model accuracy and tree size. Defining this tradeoff is difficult as it is subjective and specific to each dataset. Therefore, we included a feature in our user interface that allows users to try different pruning steps and report the impact on accuracy. These steps are incrementally pruning 10% of the nodes in the losslessly pruned decision tree.

**Computing explanation scores.**    Similar to existing explanation methods, we can also use DT+GNN to compute importance scores for a node. We accumulate these scores layer wise. In the input layer, every node is its own explanation. In each DT+GNN layer, we compute Tree-Shap values [34] for every decision tree feature. We then compute importance updates for every decision tree feature weighted by this Tree-Shap value independently (unused features have a value of 0). If the node uses a state feature (as in Figure 3a) then we give importance to the node itself. If we use a message feature (as in Figure 3b), we distribute the importance evenly between all neighbors in this state. Last, if we use a delta feature (as in Figure 3c), we distribute positive importance between all neighbors in the majority class and also negative importance between all neighbors in the minority class. Last, we normalize the scores back to sum up to 1. In the decoder layer, we employ skip connections to also consider intermediate states (for node classification) or intermediate pooled node counts (for graph classification). We provide a formal description in Appendix B.

## 4    EXPERIMENTS

### 4.1    EXPERIMENT SETUP

**Datasets.**    We conduct experiments on two different types of datasets. First, we run DT+GNN on synthetic GNN explanation benchmarks introduced by the previous works. We use the Infection and Negative Evidence benchmarks from Faber et al. [18], The BA-Shapes, Tree-Cycle, and Tree-Grid benchmarks from Ying et al. [59], and the BA-2Motifs dataset from Luo et al. [35]. For all of these datasets, we know why a graph or a node should be in a particular class. This allows us to verify that the model makes correct decisions and we compute correct explanations. Secondly, we experiment with the following real-world datasets: MUTAG [13]; BBBP [54]; Mutagenicity [29]; PROTEINS, REDDIT-BINARY, IMDB-BINARY, and COLLAB [6]. We provide more information for all datasets, such as statistics, descriptions, and examples in Appendix C. Note that all datasets except COLLAB are small enough to train on a few commodity CPUs. For example, training the PROTEINS dataset for one seed on a laptop would take 5 minutes for the full 1500 epochs, a few seconds for the tree distillation, and $1 - 2$ minutes for tree pruning. The larger REDDIT-BINARY takes around one hour to train Diff-DT+GNN (if it uses all epochs) few seconds for distilling trees and around 10 minutes for pruning. Computing lossy pruning takes a comparable amount of time to lossless pruning.

**Training and Hyperparameters.**    We follow the same training scheme for all datasets similar to existing works [55]. We do a $10-$fold cross-validation of the data with different splits and train both DT+GNN and a baseline GIN architecture. Both GNNs use a $2-$ layer MLP for the update function, with batch normalization [26] and ReLu[37] in between the two linear layers. We use 5 layers of graph convolution. GIN uses a hidden dimension of 16, DT+GNN uses a state space of 10. We also further divide the training set for DT+GNN to keep a holdout set for pruning decision trees. After we train Diff-DT+GNN with Gradient Descent we distill the MLPs inside into decision trees. Each tree is limited to having a maximum of 100 nodes. The GNNs are trained on the training set for 1500 epochs, allowing early stopping on the validation loss with patience of 100. Each split uses early stopping on the validation score, the results are averaged over the 10 splits.

DT+GNN explainability allows us to see if DT+GNN uses all layers and available states. When we see unused states in the decision trees and that layers are skipped, we retrain DT+GNN with the number of layers and states that were actually used. The retrained model does not improve accuracy but it is smaller and more interpretable. We show these in Table 1b. A full model is used for GIN.

## 4.2 QUANTITATIVE RESULTS

**DT+GNN performs comparably to GIN.** First, we investigate the two assumptions that (i) Diff-DT+GNN matches the performance of GIN and (ii) that converting from Diff-DT+GNN to DT+GNN also comes with little loss in accuracy. We further investigate how pruning impacts DT+GNN accuracy. In Table 1a we report the average test set accuracy for a GIN-based GNN, Diff-DT+GNN, DT+GNN with no pruning, and the lossless version of our pruning method.

We find that DT+GNN performs almost identically to GIN. The model simplifications which increase explainability do not decrease accuracy. We observe, that tree pruning can even have a **positive** effect on test accuracy compared to non-pruned DT+GNN. This is likely due to the regularization induced by the pruning procedure.

| Dataset | GIN | DT+GNN | | | Hyperparameters | |
| | | Differentiable | No pruning | Lossless pruning | Layers | States |
|---|---|---|---|---|---|---|
| Infection | 0.98±0.04 | 1.00±0.00 | 1.00±0.00 | 1.00±0.00 | 5 | 6 |
| Negative | 1.00±0.00 | 1.00±0.00 | 1.00±0.00 | 1.00±0.00 | 1 | 3 |
| BA-Shapes | 0.97±0.02 | 1.00±0.01 | 0.99±0.01 | 0.99±0.01 | 5 | 5 |
| Tree-Cycles | 1.00±0.00 | 1.00±0.00 | 1.00±0.00 | 1.00±0.00 | 5 | 5 |
| Tree-Grid | 1.00±0.01 | 0.99±0.01 | 0.99±0.01 | 0.99±0.01 | 5 | 5 |
| BA-2Motifs | 1.00±0.00 | 1.00±0.00 | 1.00±0.00 | 1.00±0.00 | 4 | 6 |
| MUTAG | 0.88±0.05 | 0.88±0.06 | 0.88±0.06 | 0.85±0.08 | 4 | 6 |
| Mutagenicity | 0.81±0.02 | 0.79±0.02 | 0.75±0.02 | 0.74±0.02 | 3 | 8 |
| BBBP | 0.81±0.04 | 0.83±0.03 | 0.82±0.03 | 0.83±0.03 | 3 | 5 |
| PROTEINS | 0.70±0.03 | 0.71±0.02 | 0.71±0.04 | 0.71±0.04 | 3 | 5 |
| IMDB-B | 0.69±0.04 | 0.70±0.05 | 0.69±0.03 | 0.69±0.04 | 3 | 5 |
| REDDIT-B | 0.87±0.10 | 0.90±0.03 | 0.88±0.03 | 0.87±0.04 | 2 | 5 |
| COLLAB | 0.72±0.01 | 0.70±0.02 | 0.69±0.02 | 0.69±0.02 | 3 | 8 |

(a)  (b)

Table 1: a) Test set accuracies using different GNN layers. All methods perform virtually the same for all datasets. This shows that DT+GNN layers match the expressiveness of GIN in practice. (b) DT+GNN hyperparameters found through tree inspection.

**DT+GNN produces competitive explanations.** We further evaluate how good the explanations of DT+GNN are. In line with previous work, we measure if DT+GNN finds the nodes we know to be the correct explanation in the synthetic datasets Infection, Saturation, BA-Shapes, Tree-Cycles, and Tree-Grid. For example, we know that the correct explanation for a node in the Infection dataset is the shortest path from this node to an infected node. As another example, the correct explanation for every node in the house in the BA-Shapes dataset are all nodes in the house. Table 2 shows the explanation accuracy for DT+GNN and some baseline methods. We will inspect the decision process of DT+GNN on the Tree-Cycles dataset in Figure 4.3 and show that the low explanation score is due to a defect in the dataset. The decision process of DT+GNN however is correct. Similarly, we demonstrate a defect in the Tree-Grid dataset in Figure D.3. We also provide the decision process for the BA-Shapes dataset in Figure D.2. Being able to discover such defects is a benefit of the full explainability of DT+GNN.In contrast, we cannot judge whether the shortcomings for the Gradient method are because of the method or because the underlying GNN picks up on the defects.

**Pruning significantly reduces the decision tree sizes.** Secondly, we examine the effectiveness of our pruning method. We compare the tree sizes before pruning, after lossless pruning, and after lossy pruning. We measure tree size as the sum of decision nodes over all trees. Additionally, we verify the effectiveness of using our pruning criterion for reduced error pruning and compare it against simpler setups of using only the training or validation set for pruning. We report the tree sizes and test set accuracies for all configurations in Table 3.

| Method | Infection | Saturation | BA-Shapes | Tree-Cycles | Tree-Grid |
|---|---|---|---|---|---|
| Gradient | 1.00±0.00 | 1.00±0.00 | 0.882 | 0.905 | 0.667 |
| GNNExplainer | 0.32±0.09 | 0.32±0.05 | 0.925 | 0.948 | 0.875 |
| PGMExplainer | 0.38±0.06 | 0.01±0.01 | 0.965 | 0.968 | 0.892 |
| DT+GNN | 0.95±0.02 | 1.00±0.00 | 0.94±0.02 | 0.84±0.02 | 0.927±0.01 |

Table 2: Overlap of identified explanation to explanation ground truth. Numbers for GNNExplainer and PGMExplainer are taken from Ying et al. [59], Vu & Thai [51], and Faber et al. [18].

We can see that the reduced error pruning leads to an impressive drop in the number of nodes required in the decision trees. On average, we can prune about 62% of nodes in synthetic datasets and even around 84% of nodes in real-world datasets without a loss in accuracy. If we accept small drops, in accuracy we can even prune a total of 68% and 87% of nodes in synthetic and real-world datasets, respectively. If we compare the different approaches for reduced error pruning, we can see that our proposed approach of using both training and validation accuracy performs the best. As expected, pruning only on the validation set tends to overprune the trees: Trees become even smaller but there is also a larger drop in accuracy, especially in the real-world datasets. Using the training set leads to underpruning, there is no drop in accuracy but decision trees for real-world graphs tend to stay large.

| | No pruning | | REP Training | | REP Validation | | REP Ours | | REP Lossy | |
|---|---|---|---|---|---|---|---|---|---|---|
| Dataset | Accuracy | Size | Accuracy | Size | Accuracy | Size | Accuracy | Size | Accuracy | Size |
| Infection | 1.00±0.00 | 205±56 | 1.00±0.00 | 26±2 | 1.00±0.00 | 25±2 | 1.00±0.00 | 26±2 | 0.98±0.01 | 17±2 |
| Negative | 1.00±0.00 | 18±14 | 1.00±0.00 | 5±0 | 1.00±0.00 | 5±0 | 1.00±0.00 | 5±0 | 1.00±0.00 | 4±0 |
| BA-Shapes | 0.99±0.01 | 30±10 | 0.99±0.01 | 21±5 | 0.97±0.03 | 15±4 | 0.99±0.01 | 21±5 | 0.98±0.04 | 17±4 |
| Tree-Cycles | 1.00±0.00 | 19±5 | 1.00±0.00 | 11±3 | 0.99±0.02 | 9±2 | 1.00±0.00 | 11±3 | 0.99±0.01 | 9±3 |
| Tree-Grid | 0.99±0.01 | 30±13 | 0.99±0.01 | 17±8 | 0.99±0.01 | 13±4 | 0.99±0.01 | 15±8 | 0.99±0.01 | 15±8 |
| BA-2Motifs | 1.00±0.00 | 141±43 | 1.00±0.00 | 12±3 | 1.00±0.01 | 11±3 | 1.00±0.00 | 13±4 | 1.00±0.00 | 13±4 |
| MUTAG | 0.88±0.06 | 59±27 | 0.86±0.08 | 19±17 | 0.83±0.07 | 7±6 | 0.85±0.08 | 18±16 | 0.85±0.08 | 18±16 |
| Mutagenicity | 0.75±0.02 | 375±13 | 0.76±0.02 | 154±19 | 0.73±0.01 | 56±16 | 0.74±0.02 | 91±36 | 0.73±0.02 | 50±19 |
| BBBP | 0.82±0.03 | 366±53 | 0.84±0.02 | 88±52 | 0.79±0.04 | 8±10 | 0.83±0.03 | 46±27 | 0.82±0.03 | 31±18 |
| PROTEINS | 0.71±0.04 | 206±90 | 0.72±0.03 | 12±13 | 0.70±0.04 | 8±6 | 0.71±0.04 | 9±6 | 0.71±0.04 | 9±6 |
| IMDB-B | 0.69±0.03 | 218±32 | 0.69±0.04 | 20±9 | 0.66±0.06 | 16±6 | 0.69±0.04 | 29±9 | 0.69±0.04 | 29±9 |
| REDDIT-B | 0.88±0.03 | 248±28 | 0.88±0.02 | 53±14 | 0.85±0.04 | 28±8 | 0.87±0.04 | 49±21 | 0.87±0.04 | 38±15 |
| COLLAB | 0.69±0.02 | 301±1 | 0.70±0.02 | 36±15 | 0.67±0.03 | 22±12 | 0.69±0.02 | 30±18 | 0.68±0.02 | 21±12 |

Table 3: Running reduced error pruning (REP) on different pruning sets. Lossy pruning prunes the nodes with least loss in accuracy up to a manually chosen threshold.

### 4.3 QUALITATIVE RESULTS

**Bias Terms.** Bias terms are often overlooked but can be problematic for importance explanations [52]. When a GNN uses bias terms to predict a class, nothing of the input was used so nothing should be important [18]. We observe this in the BA-2Motifs dataset. In this dataset, a GNN needs to predict if a given graph contains a house structure or a cycle. Let us start by looking at the output layer of DT+GNN in Figure 4a. If there are 5 or more nodes that have state 3 in layer 3 we classify the graph as a house. What does state 3 in layer 3 encode? We will fully investigate this dataset in Appendix D.1 and will find that this state is given to nodes in a house. Figure 4b shows the outputs for layer 3. (State 3 is green and the house is on the left). Since no attempt is made to understand cycles, the explanations for cycles also do not make sense (Figure 4c, cycle is on the left). This shows that low explanation scores on this dataset can be misleading.

**Surplus ground truth.** Faber et al. [18] discuss that having more ground truth evidence than necessary can also cause problems with explanations. This problem can be observed on the Tree-Cycle dataset. For example, DT+GNN only looks at the second layer to make final predictions (see Figure 5a). At this stage, no node could even see the whole cycle which has a diameter of 3 — yet DT+GNN has virtually perfect accuracy. The second layer(Figure 5b assigns the cycle state 1 to nodes that have neighbors in state 3. In turn, the first layer (Figure 5c) assigns nodes to state 3 that have a degree of 2. In other words, cycle nodes are connected to degree 2 nodes. Since the remaining graph is a balanced binary tree no other node except the root has degree 2. No node looking further than its 2-hop neighborhood means the cycle node that is 3 hops away is never considered entirely. Therefore

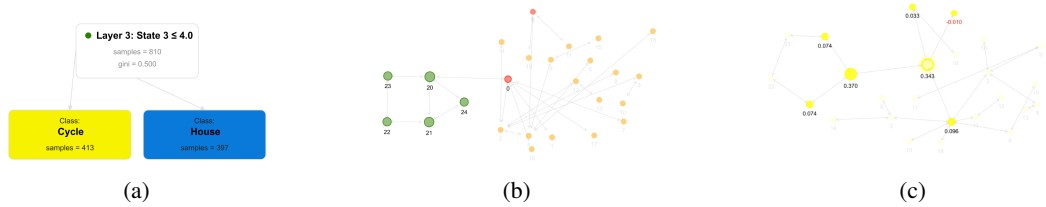

Figure 4: DT+GNN uses bias terms to solve the BA-2Motifs dataset. (a) The decoder layer only learns what a house is, the cycle is "not" a house (b) The output states of layer 3 used by the output layer. State 3 (green) is given to nodes in the house. The model learns no notion of cycles. (c) Explanation scores for cycles are not correct.

DT+GNN attributes no importance to this node which we argue is correct. This finding is consistent with the explanation accuracy score in Table 2 where DT+GNN scores very close to $\frac{5}{6}$.

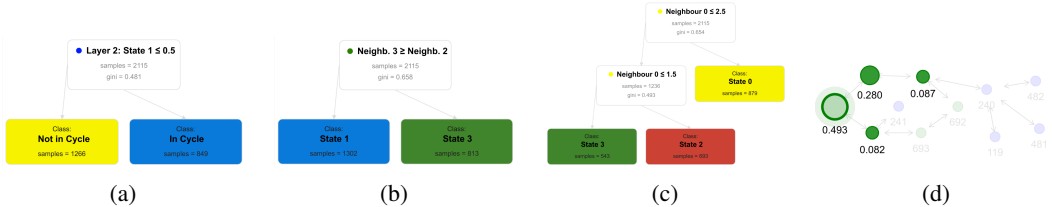

Figure 5: DT+GNN solves the Tree-Cycle dataset. (a) Final predictions require state 1 from layer 1. b) Understanding state 1 layer 1: Nodes need to have neighbors in state 0. c) Understanding this state 0: Nodes that have a degree of 2. d) Importance scores for this dataset. We can only understand through the decision process why not all cycles are important as we might have expected.

**Simple solution for MUTAG** We found inconsistencies regarding the MUTAG and Mutagenicity datasets. Several works [14; 35; 59; 60; 61] use the Mutagenicity dataset but call it MUTAG. This can lead to errors when finding explanations. Previous methods[14; 17; 59; 61] consider $NO_2$ subgraphs as correct explanations [13]. However, this explanation is not correct in MUTAG since all graphs have this group. We find an alternative, reasonably easy solution: A graph is mutagenic if it has at least 12 non-$O$ atoms and at least 8 of the following atoms: $O$ atoms, atoms connected to dominantly $O$s, atoms connected to at least three other atoms. We explain this solution in more detail in Appendix D.4.

## 5 CONCLUSION

In this paper, we presented DT+GNN which is a fully explainable graph learning method. Full explainability means that we can follow the decision process of DT+GNN and observe how information is used in every layer, yielding an inherently explainable and understandable model. Under the hood, DT+GNN employs decision trees to achieve its explainability. We experimentally verify that the slightly weaker GNN layers used do not have a large negative impact on the accuracy and that the employed tree pruning methods are very effective. We showcase some examples of how DT+GNN can help to gain insights into different GNN problems. Moreover, we also provide a user interface that allows for easy and interactive exploration of the data and decision process with DT+GNN. We believe that DT+GNN will help improve our understanding of GNNs and graph learning tasks they are used for. We hope that this leads to increased transparency of predictions made by GNNs. This will be crucial in the adoption of GNNs in more critical domains such as medicine and should help avoid models that make biased or discriminatory decisions. Similar to the MUTAG or PROTEINS examples, this transparency can also help experts in various domains to better understand their datasets and improve their approaches. As a limitation, we have observed, that DT+GNN struggles with datasets that have many node input features such as Cora [49] or OGB-ArXiv [23]. We discuss this in more detail in Appendix E.

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

## A  USING THE TOOL

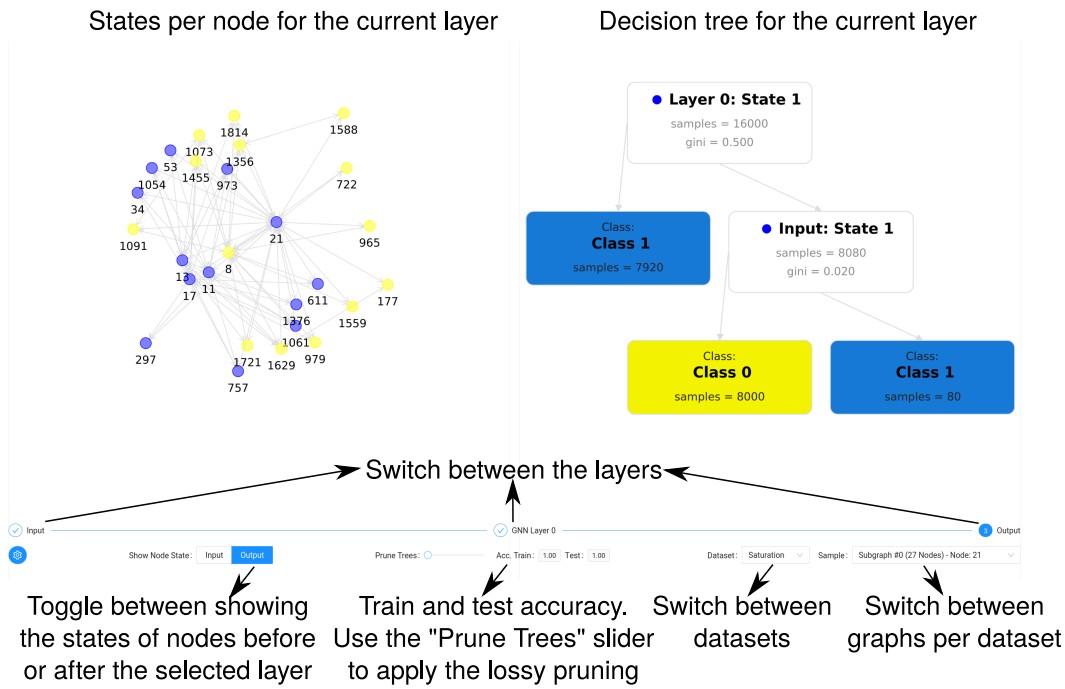

Figure 6: Initial page for the web tool. We can see the decision trees for DT+GNN per dataset and which node for a graph is in what state. We can switch layers, graphs and datasets. We can also see the test accuracy for the current setting and choose an amount of lossy pruning. with the slider.

A example instance of the tool is deployed and available via Netlify[2] and can be accessed under the link `https://interpretable-gnn.netlify.app/`. The supplementary material also contains code to host the interface yourself, in case you want to try variations of DT+GNN. In the backend, we use PyTorch [38][3] and PyTorch Geometric [19][4] to train DT+GNN and SKLearn[39][5] to train the decision trees.

The tool is built with React, in particular the Ant Design library.[6] We visualize graphs with the Graphin library.[7] The interface is a single page that will look similar to Figure 6.

The largest part of the interface is taken by two different panels at the top. In the right panel, you can see the decision tree for the currently selected layer. The trees use the three branching options from Figure 3. In the interface, evaluating the branching to true means taking the left path (this is opposite to Figure 3, which we will flip). In the left panel, you can see an example graph and which nodes end up in which state after this layer (in the bottom left you can toggle to see the input states instead). This panel does not show the full graph (most graphs in the datasets are prohibitively large) but an excerpt around an interesting region. Directly below these two graphics, you have the option to switch between layers by clicking on the respective bubble.

In the bottom right, you can switch to a different graph in the same dataset or to a different dataset. In the centre, you can see the accuracy of DT+GNN with the displayed layers. The slider allows to apply the lossy pruning from Section 3.2 and the accuracy values update to the selected pruning level.

---

[2]`https://netlify.com/`

[3]`https://github.com/pytorch/pytorch`

[4]`https://github.com/pyg-team/pytorch_geometric`

[5]`https://github.com/scikit-learn/scikit-learn`

[6]`https://github.com/ant-design/ant-design/`

[7]`https://github.com/antvis/Graphin`

The interface also allows us to examine a single node more closely by clicking on it (see Figure 7; here we clicked the blue node on the very right). Selecting reveals two things: In the graph panel, you can see the explanation scores from Section B for this node in this layer. In the tree panel, you can see the decision path in the tree for this node. This is particularly helpful if multiple leaves in the tree would lead to the same output state as in this example.

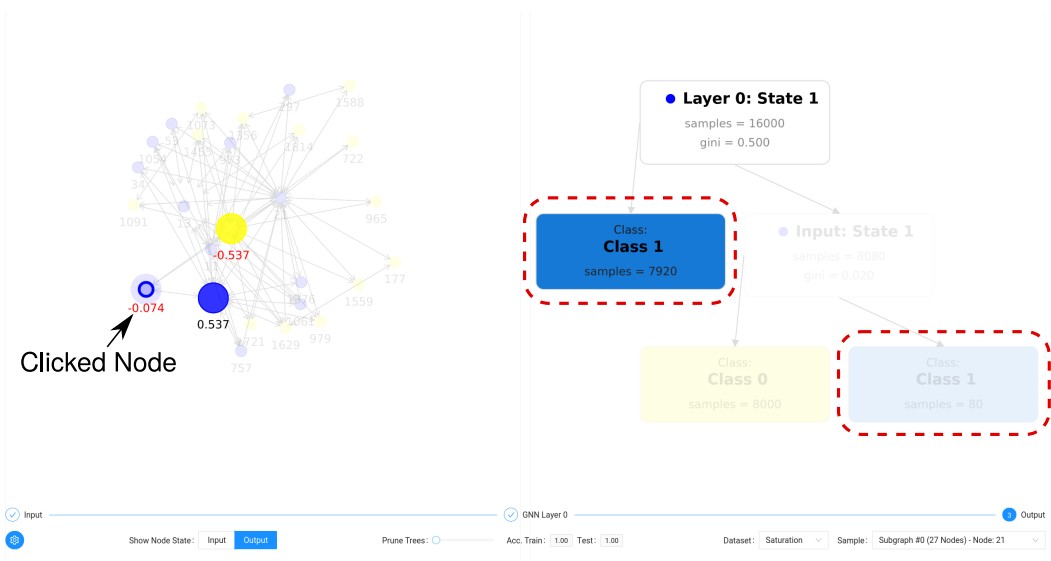

Figure 7: Interface when clicking on a node for closer examination. We can see node-level importance scores for this node on the left and the taken decision path on the right. Two paths end in the blue state, shown by the red boxes. The path the node takes is highlighted, the other path is blurred out.

# B   GENERATING EXPLANATIONS

In this section, we describe in detail how we can use DT+GNN to derive importance scores for the classification of a single node/graph. As in existing explanation methods these scores form a heatmap over all nodes to identify important inputs. While these scores are not as rich for understanding the model as inspecting the actual decision process of DT+GNN, they can visually help in guiding the user.

Formally we are going to compute scores of the form $\mathbb{R}^{N \times S \times N}$ where $N$ is the number of nodes and $S$ the number of categorical states. We assume for simplicity that every layer has the same number of states. For one node $u$ and one state $s$ the explanation $e(u, s)$ is a real-valued vector that assigns every other node $v$ an importance how much $v$ contributes to $u$ being in state $s$. We accumulate the importance over layers.

Importance for every node $u$ for the encoder layer are initialized as $e(u, v) = \mathbb{1}_u$ for every state $v$, where $\mathbb{1}_u$ is a vector that is 1 at the index of $u$ and 0 everywhere else. In other words, every node is its own explanation after the encoder.

To compute the explanation update for node $u$ in a DT+GNN layer, we investigate its decision tree. First, we compute the Tree-Shap values for $u$ in the decision tree. These values reveal how important each decision feature in the tree are for predicting $u$; a value of 0 corresponds to an unused decision. Depending on the type of decision feature — a state feature, a message features, or a delta feature (see possible cases in Figure 3) — we will add explanation to nodes differently. We handle each decision feature independently and weigh it with it's Tree-Shap value.

**State features.** There are $S$ possible state features that can each lead to $S$ different new states. This yields $S \times S$ Tree-Shap values that we denote with $\tau_S(s, s')$ To compute explanations we additionally require the indicator variable $\text{sign}(s)$ that is 1 if $u$ is in state $s$ at the start of the layer, and $-1$ otherwise. This indicator allows us to measure negative evidence that $u$ is *not* in a certain state. The "propagation" of state features is then easy since all importance stays with the node.

$$\sigma(u, s') = \sum_{s \in S} \tau_S(s, s') \cdot e(u, s) \cdot sign(s)$$

**Message features.** There are also $S$ message features that can lead to $S$ different states, thus we have $S \times S$ Tree-Shap values $\tau_M(s, s')$. Computing explanations for a neighbor feature gives each neighbor in the state $s$ importance, normalized by the number of neighbors. Let $N(s)$ denote $u$'s neighbors in state $s$:

$$\mu(u, s') = \sum_{s \in S} \tau_M(s, s') \cdot \sum_{v \in N(s)} \frac{e(v, s)}{|N(s)|}.$$

**Delta features.** We have $S^2 - S$ delta features where $(s, s')$ encodes the feature that there are more neighbors in state $s$ than neighbors in $s'$. Here we use the Tree-Shap values $\tau_\Delta(s, s', s'')$. We also need the indicators variable $(\mathbb{1}_{>(s,s')})$ that are 1 if indeed more neighbors are in state $s$ rather than $s'$ and 1 if not. Now, explanation for delta features is similar to that of neighborhood features, where the majority class contribution is positive and the minority class contribution is negative:

$$\delta(u, s'') = \sum_{s \in S} \sum_{s' \neq s \in S} \tau_\Delta(s, s', s'') \frac{\sum_{v \in N(s)} e(v, s) - \sum_{v \in N(s')} e(v, s')}{|N(s)| + |N(s')|} \cdot \mathbb{1}_{>(s,s')}.$$

These explanations are added to those of the previous layers:

$$e(u, s) = e(us, s) + \sigma(u, s) + \mu(u, s) + \delta(u, s)$$

**Decoder layer** The decoder layer is slightly special since it uses skip connections. For node classification, we directly concatenate all intermediate features and use the same computation scheme to compute the final explanations. For graph classification we additionally need to pool the nodes. We do this layer-wise and supply the decoder layer with per-layer node counts per state. The decoder can then use counting and comparison features similar to $M$ and $\Delta$ features in the DT+GNN layers. The only difference is that instead of propagating the explanation to neighbors, we now need to propagate it to all of the nodes in the graph that were in the corresponding states.

## C DATASETS

### C.1 SYNTHETIC DATASETS

- **Infection** [18] is a synthetic node classification dataset. This dataset consists of randomly generated directed graphs, where each node can be healthy or infected. The classification task predicts the length of the shortest directed path from an infected node.
- **Negative Evidence** [18] is a synthetic node classification dataset. A random graph with ten red nodes, ten blue nodes, and 1980 white nodes is created. The task is to determine whether the white nodes have more red or blue neighbours.
- **BA Shapes** [59] is a synthetic node classification dataset. Each graph contains a Barabasi-Albert (BA) base graph and several house-like motifs attached to random nodes of the base graph. The node labels are determined by the node's position in the house motif or base graph.
- **Tree Cycle** [59] is a synthetic node classification dataset. Each graph contains an 8-level balanced binary tree and a six-node cycle motif attached to random nodes of the tree. The classification task predicts whether the nodes are part of the motif or tree.
- **Tree Grid** [59] is a synthetic node classification dataset. Each graph contains an 8-level balanced binary tree and a 3-by-3 grid motif attached to random nodes of the tree. The classification task predicts whether the nodes are part of the motif or the tree.
- **BA 2Motifs** [35] is a synthetic graph classification dataset. Barabasi-Albert graphs are used as the base graph. Half of the graphs have a house-like motif attached to a random node, and the other half have a five-node cycle. The prediction task is to classify each graph, whether it contains a house or a cycle.

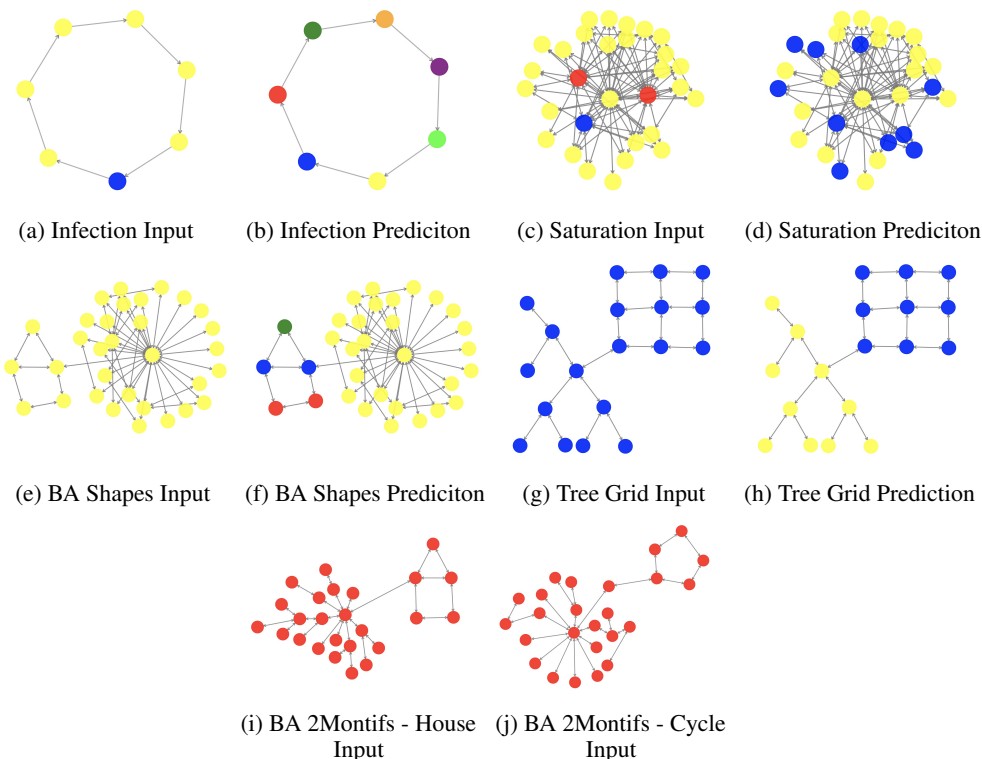

(a) Infection Input    (b) Infection Prediciton    (c) Saturation Input    (d) Saturation Prediciton

(e) BA Shapes Input    (f) BA Shapes Prediciton    (g) Tree Grid Input    (h) Tree Grid Prediction

(i) BA 2Montifs - House Input    (j) BA 2Montifs - Cycle Input

Figure 8: Synthetic Benchmarks - Example Graphs

### C.2 REAL-WORLD DATASETS

- **MUTAG** [13] is a molecule graph classification dataset. Each graph represents a nitroaromatic compound, and the goal is to predict its mutagenicity in Salmonella typhimurium. Mutagenicity is

| Dataset | Graphs | Classes | Avg. Nodes | Avg. Edges | Features |
|---|---|---|---|---|---|
| Infection | 1 | 7 | 1000 | 3973 | 2 |
| Negative Evidence | 1 | 2 | 2000 | 102394 | 3 |
| BA Shapes | 1 | 4 | 700 | 4110 | 0 |
| Tree Cycle | 1 | 2 | 871 | 1942 | 0 |
| Tree Grid | 1 | 2 | 1231 | 3130 | 0 |
| BA 2Motifs | 1000 | 2 | 25 | 50.96 | 0 |

Table 4: Statistics of Synthetic Datasets

the ability of a compound to change the genetic material permanently, usually DNA, in an organism and therefore increase the frequency of mutations. The nodes in the graph represent atoms and are labeled by atom type. The edges represent bonds between atoms.

- **Mutagenicity** [29] is a molecule graph classification dataset. Each graph represents the chemical compound of a drug, and the goal is to predict its mutagenicity. The nodes in the graph represent atoms and are labeled by atom type. The edges represent bonds between atoms.

- **BBBP** [54] is a molecule graph classification dataset. Each graph represents the chemical compound of a drug, and the goal is to predict its blood-brain barrier permeability. The nodes in the graph represent atoms and are labeled by atom type. The edges represent bonds between atoms.

- **PROTEINS** [6] is a protein graph classification dataset. Each graph represents a protein that is classified as an enzyme or not and enzyme. Nodes represent the amino acids, and an edge connects two nodes if they are less than 6 Angstroms apart.

- **REDDIT BINARY** [6] is a social graph classification dataset. Each graph represents the comment thread of a post on a subreddit. Nodes in the graph represent users, and there is an edge between users if one responded to at least one of the other's comments. A graph is labeled according to whether it belongs to a question/answer-based or a discussion-based subreddit.

- **IMDB BINARY** [6] is a social graph classification dataset. Each graph represents the ego network of an actor/actress. In each graph, nodes represent actors/actresses, and there is an edge between them if they appear in the same film. A graph is labeled according to whether the actor/actress belongs to the Action or Romance genre.

- **COLLAB** [6] is a social graph classification dataset. A graph represents a researcher's ego network. The researcher and their collaborators are nodes, and an edge indicates collaboration between two researchers. A graph is labeled according to whether the researcher belongs to the field of high-energy physics, condensed matter physics, or astrophysics.

- **Cora**, **CiteSeer**, and **PubMed** are popular citation networks [58]. Nodes are papers and citations are edges. Nodes contain features that represent words of their contents and are labeled by sub-fields.

| Dataset | Graphs | Classes | Avg. Nodes | Avg. Edges | Features |
|---|---|---|---|---|---|
| MUTAG | 188 | 2 | 17.93 | 39.59 | 7 |
| Mutagenicity | 4337 | 2 | 30.32 | 61.54 | 14 |
| BBBP | 2039 | 2 | 24.06 | 51.91 | 9 |
| PROTEINS | 1113 | 2 | 39.06 | 145.63 | 3 |
| REDDIT BINARY | 2000 | 2 | 429.63 | 995.51 | 0 |
| IMDB BINARY | 1000 | 2 | 19.77 | 193.06 | 0 |
| COLLAB | 5000 | 3 | 74.49 | 4914.43 | 0 |
| Cora | 1 | 7 | 2485 | 5069 | 1433 |
| CiteSeer | 1 | 6 | 2110 | 2668 | 3703 |
| PubMed | 1 | 3 | 19717 | 44324 | 500 |

Table 5: Statistics of Real-World Datasets

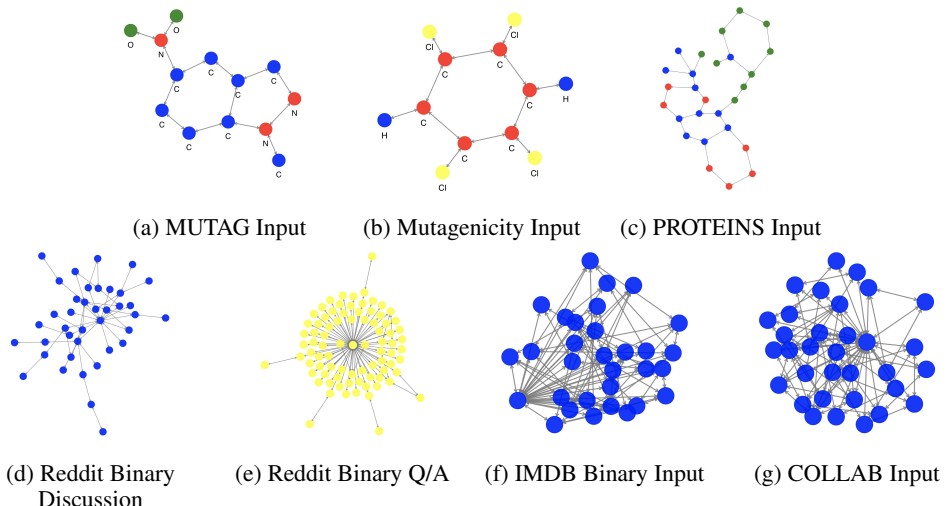

(a) MUTAG Input     (b) Mutagenicity Input     (c) PROTEINS Input

(d) Reddit Binary Discussion     (e) Reddit Binary Q/A     (f) IMDB Binary Input     (g) COLLAB Input

Figure 9: Real-world benchmarks - Example graphs

# D    MORE QUALITATIVE EXPERIMENTS

## D.1    BA-2MOTIFS

Let us know look in detail at the decision process of DT+GNN in the BA-2Mofits dataset, that we mentioned in Section 4.3. Basically this dataset only identifies one class (houses) while solving the other class (cycles) via bias terms. We now investigate the decision process to find houses. This time, we present an alternative approach to investigate DT+GNN that reminds of dynamic programming: We are going to go through layers from front to back and note what the state per layer encode.

The first layer (Figure 10a) splits the nodes by their degree: degree 1 nodes receive state 5, degree 2 nodes receive state 2, other nodes receive state 4. The second layer (Figure 10b) is not required for computation and shuffles states around: degree 1 nodes receive state 3, degree 2 nodes receive state 0, other nodes receive state 5. In the third layer (Figure 10c) nodes in state 3 (degree 1 nodes) receive state 3, nodes with more degree 1 than "two or more" neighbors receive state 5. Other nodes (including those in the house that do not have any degree 1 neighbor) receive "house candidate" state 1. Finally (Figure 10d), only nodes in the house receive the "house" state 3 by requiring at least two house candidate neighbors.

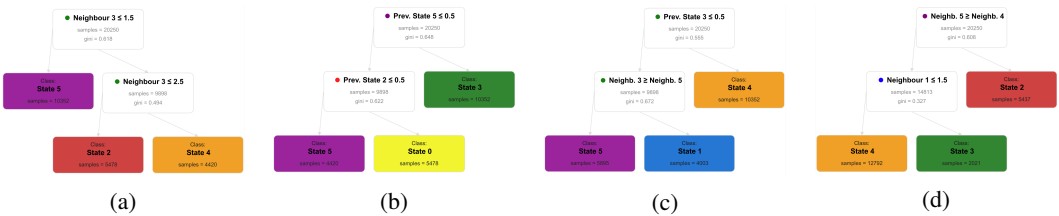

|                (a)                |                (b)                |                (c)                |                (d)                |

Figure 10: Forward analysis of DT+GNN for the BA-2Motifs dataset. (a) and (b) separate degree 1 from degree 2 from other nodes. (c) Move nodes that do not have many degree 1 neighbors in state 1 (this includes all house nodes which have no such neighbors). Let's call these nodes "house candidates" (d) House nodes have at least 2 house candidate neighbors.

## D.2    BA-SHAPES

Let us now look at the related problem and identify houses and also the position of each node (top, middle, bottom) in the house. The first layer (Figure 11a) identifies most of the house by finding degree 3 or lower nodes in state 0 versus "high-degree" nodes in state 1 (the rest of the graph has high degrees). The second layer (Figure 11b) consolidates the house as nodes that do not have 4 or more high-degree neighbors. House nodes receive state 2. The next layer's tree learnt to be unnecessarily complex, Figure 11c shows the important subtree which divides the middle part of the house from the other parts by a degree check (state 0 versus state 1). Then, we can separate the top from the bottom nodes if they have an equal neighbor (Figure 11d). DT+GNN does one extra computation step afterward since received an extra layer but we already found the house and the position of each node in it.

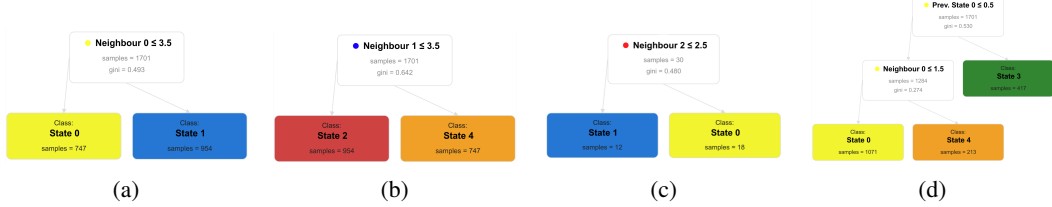

|                (a)                |                (b)                |                (c)                |                (d)                |

Figure 11: Forward analysis of DT+GNN for the BA-Shapes dataset. (a) and (b) identify the house nodes through a degree check — the outer graph is much more densely connected. (c) separate the middle part of the house (degree 3 from the others with degree 2). (d) separate the top from the bottom (another bottom node as a neighbor).

### D.3  TREE-GRID

The Tree-Grid [59] dataset is similar to the Tree-Cycles dataset we discussed in the main body of the paper. The base graph is a balanced binary tree to which we append $3 \times 3$ grids. As in the Tree-Cycles example, there are (apart from the root node) no other nodes with degree 2 which makes bootstrapping the grid discovery easier. As in the Tree-Cycles example, a GNN does not need to see the whole grid to make a prediction.

DT+GNN starts (Figure 12a with distinguishing nodes if they have degree 1 (state 2), degree 2 (state 4), otherwise state 1. The important part in Figure 12b is finding the neighbors of the degree 2 nodes. These are mapped to state 1 and transformed to state 2 in the next layer (Figure 12c). In Figure 12d, we identify the first grid nodes in state 3: such are nodes that have many neighbors that were in state 2 in the previous layer. We find the remaining grid nodes as the nodes that have 2 grid neighbors (Figure 12e). In the decoder (Figure 12e), we read out these two layers to find the whole grid.

Some nodes found their place in the grid after just three layers — importantly the corner nodes generally belong to this group. However, there exists an opposite corner node that is 4 hops away that is considered unimportant. Consequently, DT+GNN does not include this node in its explanation. This is consistent with the explanation accuracy in Table 2: DT+GNN achieves a bit more than $\frac{8}{9}$ — since some nodes use all 4 layers after all.

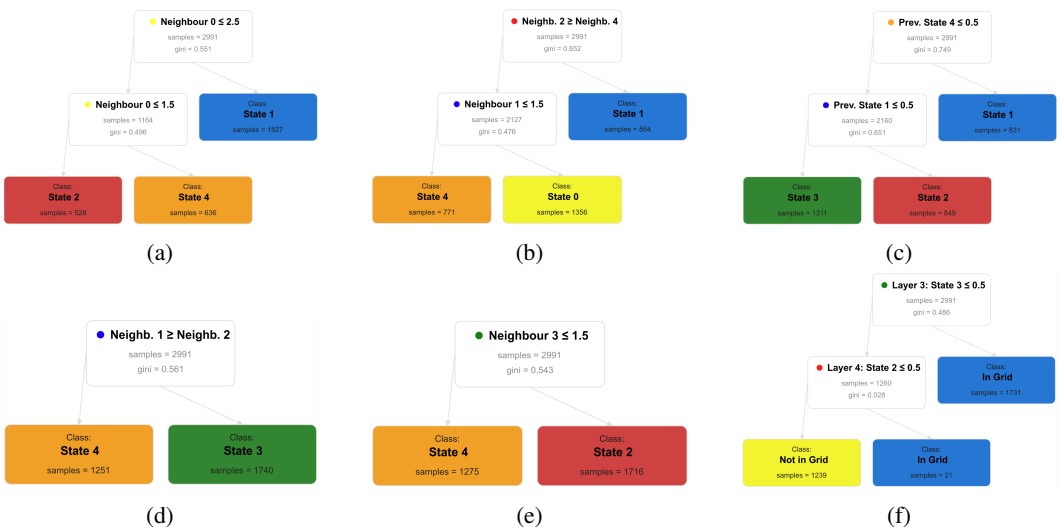

Figure 12: Forward analysis of DT+GNN for the Tree-Grid dataset. Figures (a)-(c) show the progression of node states for the first three layers. Numbers denote node IDs. (a) DT+GNN finds nodes of degree 2 in the first layer, the corners of the grid. (b) DT+GNN finds the neighbors of the corner nodes. (c) DT+GNN finds the nodes next to the previous nodes — completing the grid. (d) This plot shows the explanation scores for the top-right node. Explanations are the $2-$distance neighborhood. Importantly, the bottom-left node in the grid was never seen by the top-right node — rightfully it is not part of the explanation.

### D.4  MUTAG

As discussed in the main body, the presence of $NO_2$ groups in MUTAG [13] is not informative for mutagenicity since all graphs have such a group. We will present another decision rule based on interpreting DT+GNN's decision process here. Let us look at the output layer (Figure 13c). A graph is not mutagenic if it has less than 12 nodes in layer 2 state 3. We now recursively look up what this state is, these are nodes other than $O$. Here, DT+GNN found a solution to encode this somewhat complicated but becomes clear by looking at the state assignment per node (Figure 13a). This showcases that explanations and visual tools empower each other well. The second condition in Figure 13c concerns nodes in layer 3 state 5: if there are least 8 such nodes the graph is mutagenic otherwise it is not. Figure 13b shows the decision process for this state. Ultimately three kinds of

nodes arrive in this state: Nodes that have at least three neighbors (some $C$ atoms), $O$ atoms (the previous state was not 3, and nodes primarily connected to $O$ atoms.

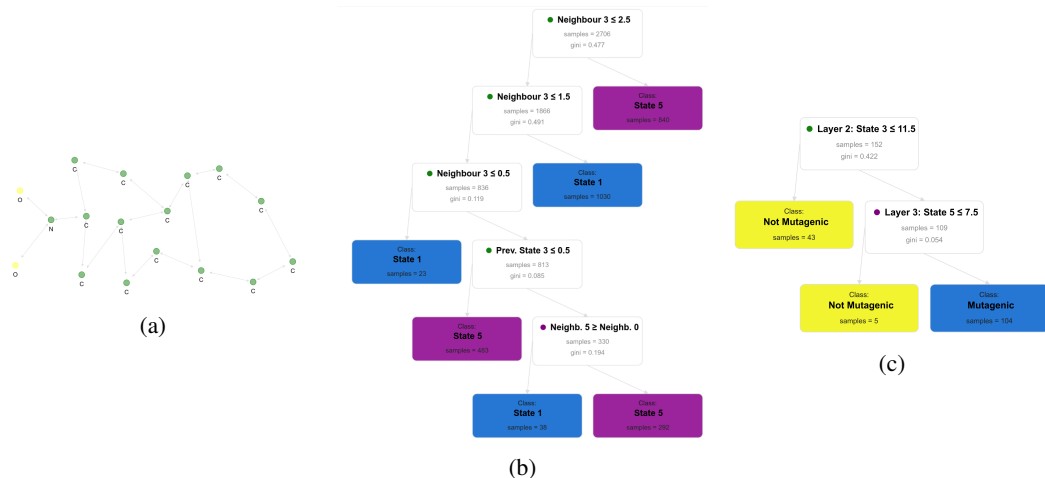

Figure 13: Decision process for DT+GNN on the MUTAG dataset. (a) For a graph to be mutagenic, we require at least $12$ layer 2 state 3 nodes and $8$ layer 3 state 5 nodes. (b) layer 2 state 3 nodes are non-$O$ atoms. (c) layer 3 state 5 nodes are (i) $O$ atoms, atoms connected to $O$s, or atoms with at least three neighboring atoms.

## D.5    REDDIT-BINARY

Last, we look at the real-world REDDIT-BINARY dataset that consists of unattributed graphs representing reddit threads. Nodes are users with edges connecting users that commented on one another. Depending on the subreddit the threads come from, graphs are labeled as "Discussion" or "Q&A".

DT+GNN starts by splitting nodes into three groups based on their degree (Figure 14c: Nodes with a degree smaller than 3 receive state 2, nodes with a degree of more than $46$ receive state 0, the intermediate nodes receive state 4. We can see in Figure 14c that degree $0$ from the second layer are crucial: The graph is "Q%A" if there are 15 or more such nodes. Figure 14b shows what this state encodes: First of all, there must be high-degree nodes (state 0 in the neighborhood). DT+GNN assigns node state 0 if they are: (i) a low-degree node (in state 2) that is connected to at least 2 high degree (state 0) nodes (ii) or other nodes that are connected with few medium-degree nodes. These two cases make sense: In a "Q&A" graph there are responding users that will a relatively much higher degree and the graphs resemble a star-like structure. There will be users that just ask a question and do not interact much else (case (i)). Case (ii) captures smaller potential "Q&A" graphs: this case looks for medium nodes where there are few other medium nodes. In these cases, these nodes are likely the answer nodes. On the other hand, in "Discussion" graph users tend to generally reply to each other, creating less of a star-like graph.

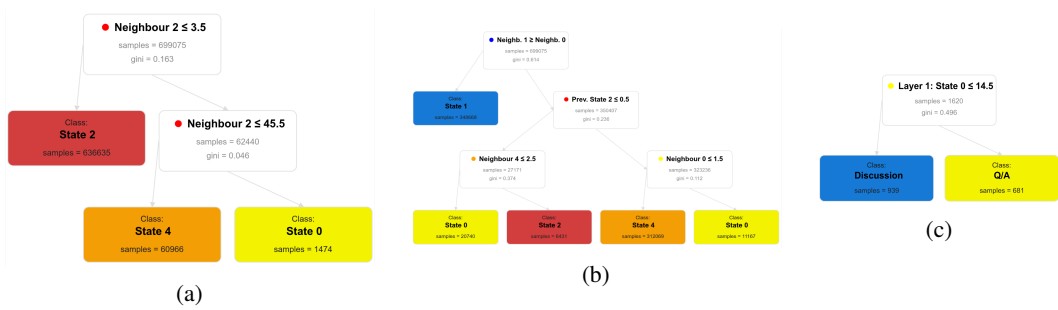

Figure 14: Decision process of DT+GNN for the REDDIT-BINARY dataset. (a) DT+GNN separates the nodes into low-degree (state 2), high-degree (state 0), and the remaining medium-degree nodes (state 4). Generally, a "Q&A" graph exhibits a star-like structure with few nodes answering questions from all others. (b) DT+GNN captures this idea by looking for nodes that are (i) low degree and connected to high-degree nodes or (ii) connected to few medium-degree nodes. (c) A graph is "Q&A" if there are at least 15 such nodes.

# E  DT+GNN ON DATASETS WITH MANY INPUT FEATURES

| Dataset | Features | GIN | DT+GNN Differentiable | No pruning | Lossless pruning |
|---|---|---|---|---|---|
| CORA | 1433 | 0.87±0.02 | 0.82±0.03 | 0.69±0.04 | 0.68±0.03 |
| CiteSeer | 3703 | 0.77±0.01 | 0.70±0.03 | 0.61±0.04 | 0.61±0.02 |
| PubMed | 500 | 0.88±0.01 | 0.87±0.01 | 0.85±0.01 | 0.85±0.01 |
| OBGN-Arxiv | 128 | 0.68±0.02* | 0.68±0.01 | 0.28±0.11 | - |

Table 6: DT+GNN results for citation datasets with high-degree counts. *Since the dataset has 40 classes, we use a state-size of 50 for DT+GNN variants and 128 wide embeddings for GIN.

In the following we want to discuss DT+GNN on high-dimensonal datasets such as Cora (1433) features. Table 6 shows a comparison of GIN, Diff-DT+GNN and DT+GNN similar to Table 1a. The results are mixed: on Pubmed, DT+GNN performs comparable to GIN, on Cora there is a small drop for Diff-DT+GNN but a significant drop when converting to trees. For CiteSeer, both Diff-DT+GNN and converting to trees cause clear drops in accuracy. We see two factors that make this dataset challenging: Large feature spaces make it harder to reduce to a categorical state. For example for the Cora dataset, the encoder needs to reduce from 1433 to 10 features. This effect increases in DT+GNN when we limit the number of leaves: Having 100 decision leaves means that a tree can have 99 decision nodes and look at most at 99 features. But already such trees are impractical to interpret. We found that even after pruning, the trees often contain long paths of depth 20 or more. The problems aggravate on the larger OBGN-Arxiv dataset: Diff-DT+GNN performs decently with a drop comparable to CiteSeer but DT+GNN drops drastically in accuracy. Furthermore, this dataset reveals scalability limits for DT+GNN's pruning method: Pruning requires the number of leaves squared many runs over the dataset and does not scale to this dataset.

Therefore we believe that handling such datasets requires a different approach. In future work, we image that these issues could be addressed through approaches such as PCA, clustering, or special MLP construction techniques [53; 46] to reduce the input space without breaking the interpretability chain before applying DT+GNN.

