# OpenReview forum: "DT+GNN: A Fully Explainable Graph Neural Network using Decision Trees"
_ICLR.cc/2023/Conference — Submitted to ICLR 2023_

### Official Review · Reviewer_9FEb · 2022-10-24

**Confidence:** 3
**Correctness:** 4
**Technical Novelty And Significance:** 3
**Empirical Novelty And Significance:** 3
**Recommendation:** 6

**Clarity, Quality, Novelty And Reproducibility:**

[1] This paper is clearly written and well organized.

[2] Most of the claims are well supported by visual examples or experimental results.

[3] The idea of distilling neural networks into trees to make them explainable have seen before, but the way to realize it in GNN seems novel to me.

[4] The code is provided for reproducibility.

**Strength And Weaknesses:**

Strength:

[1] By introducing the Decision Tree to GNN, humans can inspect and understand the decision-making of DT+GNN at every step, which is valuable for the community.

[2] Pruning these decision trees will make them easier to interpret, and small trees are efficient for inference.

[3] Experiments show DT+GNN performs almost identically to GIN in real-world datasets and produces competitive explanations in synthetic datasets.

Weaknesses:



**Summary Of The Paper:**

This paper proposed a novel Decision Tree GNN architecture, which is fully explainable. DT+GNN firstly trains a fully differentiable layer that is restricted to categorical state spaces for nodes and messages. Secondly, they distill these layers into decision trees. Finally, pruned these trees to ensure they are small and easy to interpret. DT+GNN performs comparably to GIN in synthetic and real-world datasets.


**Summary Of The Review:**

This paper proposed a novel DT-GNN for understanding decision-making at every step. Most of their claims are well supported by visual examples or experimental results. It deserves to be accepted by this conference.

---

> ### Author Response · Authors · 2022-11-13
> **Response to reviewer 9FEb**
>
> Thank you for your review. We noticed you haven’t listed any weaknesses in your review. Do you see any improvements or clarifications we could make to further increase your confidence in our work?

---

### Official Review · Reviewer_gfpK · 2022-10-24

**Confidence:** 3
**Correctness:** 2
**Technical Novelty And Significance:** 2
**Empirical Novelty And Significance:** 2
**Recommendation:** 6

**Clarity, Quality, Novelty And Reproducibility:**

Clarity:
- No major concern. Authors have added a link to their website with visualization in page 2 that helps better understanding how the proposed method works which is great.

Quality:
- No major concern.

Novelty:
- Contributions such as using stone age model, pruning the trees, representing MLP with decision tree, etc. are not individually novel and utilizing the combination of these techniques make the work slightly novel.

Reproducibility:
- No major concern.

**Strength And Weaknesses:**

Strength
- The paper is easy to read and adequate amount of explanations are provided to make the proposed concepts easier to grasp for the reader.
- Literature review looks good for the most part, except some more general recent efforts on replacing Neural Networks with DTs (e.g. Neural Networks are Decision Trees by Aytekin et. al.) that can be applicable to GNN as well.
- Experimental results are comprehensive and support the claims.

Weaknesses
- As mentioned by authors, there is a shortcoming in the proposed method's performance for domains with large number of node input features.
- There is a disconnect between studying the performance of DT+GNN and its explainability since alternative methods reported for explainability comparisons are not used in the performance comparison, therefore, one cannot get a wholistic picture of the trade-off between explainability and performance for the alternative methods.
- Experimental results do not seem to highlight significant benefits neither for performance nor for explainability.
- Also refer to Novelty section in the next box.

**Summary Of The Paper:**

The paper proposes a new Decision Tree GNN (DT+GNN) architecture to help make GNN predictions more explainable. The proposed approach uses a new layer inspired by stone age model, distills all MLPs to Decision Trees, introduces a pruning mechanism for there trees, and reports experimental results to empirically evaluate the method's performance and explainability with other alternative approaches. Experimental results show that DT+GNN has comparable performance with GIN, a GNN based method. Results also show that DT+GNN can achieve comparable explainability scores with a few other alternative methods.

**Summary Of The Review:**

The paper addresses an important topic in GNN, i.e. explainability, which is an active research area. The proposed method is explained clearly and experimental results are provided to support its benefits.

---

> ### Author Response · Authors · 2022-11-13
> **Response to reviewer gfpK - Discussion on new literature, benefits and novelty of DT+GNN**
>
> Thank you for your review
>
> > Literature review looks good for the most part, except some more general recent efforts on replacing Neural Networks with DTs (e.g. Neural Networks are Decision Trees by Aytekin et. al.) that can be applicable to GNN as well.
>
> Thanks for highlighting this paper. It has come out very recently and after the submission deadline for ICLR but complements the spirit of our work well. This work might raise the question if we need the stone age layer if we could also create a tree straight away. However, the authors show that creating trees this way can become very (exponentially) large. That makes them impossible to interpret (for the same reason, we stay with small categorical states even if it hurts expressiveness). Furthermore, their tree construction does not make the messages for message passing interpretable which further prohibits understanding of the GNN dynamics and the entire model. This is the main benefit of the stone age layers: We can decompose the trees by layer and get messages for message passing we can understand. We add this discussion in the paper in the related work section.
>
> > There is a disconnect between studying the performance of DT+GNN and its explainability since alternative methods reported for explainability comparisons are not used in the performance comparison, therefore, one cannot get a holistic picture of the trade-off between explainability and performance for the alternative methods.
>
> GIN can be seen as the performance baseline for the alternative explainability methods. Methods like GNNExplainer, PGM-Explainer or gradient methods do not make any assumptions about the model and can be used on any GNN. This appealing property also limits what insight they can give us, they produce importance scores that allow heatmap-like importance scores.
>
> DT+GNN is the only model that reveals its actual decision process, what features or neighborhood counts are used in each layer.
>
> > Experimental results do not seem to highlight significant benefits neither for performance nor for explainability.
>
> > Contributions such as using stone age model, pruning the trees, representing MLP with decision tree, etc. are not individually novel and utilizing the combination of these techniques make the work slightly novel.
>
> Actually, we would argue that *not seeing drops* as is already a win. Theoretically, DT+GNN has a lower expressive power so we are delighted to see that this is not a practical obstacle.
>
> The main benefit of DT+GNN is not to provide better importance scores. If anything, these scores are a byproduct. The real benefit of DT+GNN is that we do not only find importance scores but the full decision process. We expanded on this in the introduction with an example PROTEINS graph. With importance scores, we can see that the two yellow nodes are important. But we cannot understand why the model considers them important: because they are connected? There were more than one yellow node? There were less than 5 yellow nodes? Importance scores cannot answer such questions, we can get only a rough idea by looking at scores over many graphs.
>
> On the other hand DT+GNN gives this information by revealing the decision process. In the example, the graph is an enzyme because there are less than 3 yellow nodes. This id a new quality of explanations and possible insights. This is also what we see as the novelty in the paper — as you correctly said, most of the methods exist individually. Also, existing metrics do not capture this quality so it is hard to demonstrate in quantitative experiments. Therefore, we added many examples these new explanations for many datasets in the main body and Appendix D.

---

### Official Review · Reviewer_Sxw4 · 2022-10-24

**Confidence:** 3
**Correctness:** 4
**Technical Novelty And Significance:** 3
**Empirical Novelty And Significance:** 3
**Recommendation:** 6

**Clarity, Quality, Novelty And Reproducibility:**

* Clarity: Explain better the section of Distilling the DT+GNN. Write the formulation of the stone age model.
* Quality: yes
* Novelty: yes
* Reproducibility: do they provide their code?


**Strength And Weaknesses:**

## Strenghts:

Authors address a real-world problem.
Explanation method is simple and comprehensible.
Their method is empirically solid.

## Weaknesses

As the authors pointed out, their explainability method works only on small graph dataset, or datasets with limited number of features.
Since their method is theoretically less expressive than message-passing, it should have been interesting to provide performance analysis of their classifier (not the explainable module) for large datasets like Cora.


**Summary Of The Paper:**

This paper proposes a novel GNN architecture that uses a small categorical space for messages and states instead of traditional synchronous message passing. Moreover, after training, they replace all the MLPs in their layers with decision trees to give a fully interpretable model

**Summary Of The Review:**

The authors address a real-world issue. While their method is theoretically less expressive than message-passing, it is empirically competitive. The authors noted that their method is better suited to graph datasets with fewer features, since the complexity of the method increases with the number of features. This method is not suitable for large graph datasets.

---

> ### Author Response · Authors · 2022-11-13
> **Response to Reviewer Sxw4 - Mathematical formulations and scalability to large graphs**
>
> Thank you for your review
>
> > As the authors pointed out, their explainability method works only on small graph dataset, or datasets with limited number of features. Since their method is theoretically less expressive than message-passing, it should have been interesting to provide performance analysis of their classifier (not the explainable module) for large datasets like Cora.
>
> We experienced that the imitation for DT+GNN is the number of input features, not the size of the graph or the number of graphs in the dataset. If we understand correctly, with classifier, you mean the neural network that is restricted to a categorical space? We ran experiments on Cora, CiteSeer, Pubmed and OBGN-Arxiv. On Cora, Pubmed, and OBGN-Arxiv we find that Diff-DT+GNN does not loose to much accuracy.
>
> However, distilling from Diff-DT+GNN to DT+GNN comes with a clear drop in accuracy on all datasets except PubMed. Intuitively this makes sense as we aim to have shallow trees and fewer states in the model to keep it interpretable. Shallow trees are also inherently limited in the number of features they can process (a tree with 100 leaves can only have 99 splits on features).
>
> We expanded on this in the paper and added the experimental results in a new appendix E.
>
> The performance on the decently-large REDDIT-BINARY and COLLAB datasets suggests that both Diff-DT+GNN and DT+GNN can scale with the number of graphs and mean nodes per graph.
>
> > Clarity: Explain better the section of Distilling the DT+GNN. Write the formulation of the stone age model.
>
> We added the formulations of each layer (GIN, Diff-DT+GNN, DT+GNN) to section 3 and expanded the captions. We also rearranged section 3.1 a bit. The baseline layer is a GIN layer with $\epsilon=0$ which has the following formulation:
> $h_v^{l+1} = f_\theta (h_v^l, \sum_{w \in Nb(v)} h_w^l)$
>
> f_\theta is a parameterized function (e.g., a 2-layer MLP). The Stone-Age/Diff DT+GNN augments this by a Gumbel Softmax:
>
> $h_v^{l+1} = Gumbel (f_\theta (h_v^l, \sum_{w \in Nb(v)} h_w^l))$
>
> DT+GNN then replaces the neural block by a function $\tau$ (which is modeled by a decision tree; we can write this as):
>
> $h_v^{l+1} = \tau(h_v^l, \sum_{w \in Nb(v)} h_w^l)$
>
>
> > Reproducibility: do they provide their code?
>
> We attach the code as a .zip file in the supplementary material. The zip contains both the code to train DT+GNN and the code for the UI to show it. You can run locally, but you skip this and see the final results hosted at https://interpretable-gnn.netlify.app/.

---

### Official Review · Reviewer_6BLn · 2022-10-26

**Confidence:** 4
**Correctness:** 3
**Technical Novelty And Significance:** 3
**Empirical Novelty And Significance:** 3
**Recommendation:** 6

**Clarity, Quality, Novelty And Reproducibility:**

**Clarity**

The paper lacks clarity about the details of the DT+GNN architecture.


**Strength And Weaknesses:**

**Strengths**

1. The paper presents DT+GNN, a novel architecture that consists of a new differentiable Diff-DT+GNN layer inspired by a simplified distributed computing model known as the stone age model.
2. The paper provides an interactive user interface that can be used to explore the decision process of the DT+GNN model trained on different datasets.
3. In addition to the thresholds of the decision tree, DT+GNN can also be used to generate graph-level explanations.

**Weaknesses and Open Questions**
1. The paper lacks sufficient details about the DT+GNN model in Section 3. The readability will drastically improve if we describe the message-passing scheme of DT+GNN using mathematical equations. Further, the figure captions are unclear and not self-explanatory.
2. What is the loss of information when constructing a categorical state space of DT+GNN using $\mathcal{O}(n)$ bits to encode the information from the continuous embeddings?
3. Is there a hard threshold on the number of decision leaves per tree, which is restricted in the training process?
4. Most DT+GNN architecture use more than two layers for synthetic and real-world datasets. It is unclear whether both GIN and DT+GNN use these many layers during training. If yes, then it is very counterintuitive that GIN needs 5 layers to get high performance for small datasets like BA-Shapes and Tree-Cycles.
5. The limited number of categorical states in DT+GNN limits its applicability to large benchmark graph datasets.

**Summary Of The Paper:**

Explainability in Graph Neural Networks (GNNs) is in an upcoming research direction, where most recent works have focused on developing perturbation-, surrogate-, or prototype-based post hoc explanation methods. However, these methods fail to identify how a GNN model processes the input layer-wise. In this work, the authors combine decision trees (DT) with GNN to introduce a fully explainable new architecture DT+GNN, which builds on a novel differentiable GNN layer that is restricted to categorical state spaces for nodes and messages and, similar to existing post hoc GNN explainers, outputs node-level importance scores. Empirical results on real-world GNN benchmarks show that DT+GNN achieves on-par or better results than their vanilla counterparts.

**Summary Of The Review:**

The paper presents a novel interactive tool to understand the information flow of the fully explainable DT+GNN architecture but lacks sufficient motivation and technical details about the model.

---

> ### Author Response · Authors · 2022-11-13
> **Response to reviewer 6BLn 1/2: Main Benefit of DT+GNN, mathematical formulations**
>
> Thank you for your review.
>
> > In this work, the authors combine decision trees (DT) with GNN to introduce a fully explainable new architecture DT+GNN, which builds on a novel differentiable GNN layer that is restricted to categorical state spaces for nodes and messages and, similar to existing post hoc GNN explainers, outputs node-level importance scores.
>
> We want to emphasize that we see the main contribution of DT+GNN not as another method that can produce importance scores, but as a method that allows a new quality of explanation. Instead of having an idea what input was important, we get insight into the full decision process **why** It was important. We expanded on this in the introduction with an example graph with importance scores for the PROTEINS dataset: Importance scores can highlight the two yellow nodes but they cannot tell us if they are important because 1) they are connected versus because 2) there is more than one versus because 3) there are less than 5. But looking at the decision process with DT+GNN can tell us because there were less than 3 of such nodes.
>
> If anything, the importance scores are more a by-product in DT+GNN when following the decision process.
>
> > The paper lacks sufficient details about the DT+GNN model in Section 3. The readability will drastically improve if we describe the message-passing scheme of DT+GNN using mathematical equations. Further, the figure captions are unclear and not self-explanatory.
>
> We added the mathematical definitions for GIN, Diff-DT+GNN and DT+GNN in section 3 and revised the structure of the section a bit.
>
> In this paper we use GIN form Xu et al. _How powerful are Graph Neural Networks_ with $\epsilon=0$ which has the following formulation:
>
> $h_v^{l+1} = f_\theta (h_v^l, \sum_{w \in Nb(v)} h_w^l)$
>
> $f_\theta$ is a parameterized function (e.g., a 2-layer MLP). Diff DT+GNN augments this by a Gumbel Softmax to get categorical states:
>
> $h_v^{l+1} = Gumbel (f_\theta (h_v^l, \sum_{w \in Nb(v)} h_w^l))$
>
> DT+GNN then replaces the neural block by a function $\tau$ (which is modeled by a decision tree; we can write this as):
>
> $h_v^{l+1} = \tau(h_v^l, \sum_{w \in Nb(v)} h_w^l)$
>
> We also expanded on the captions. Please let us know if anything remains unclear.

---

> ### Author Response · Authors · 2022-11-13
> **Response to reviewer 6BLn 2/2 Details and Limitations of Categorical states and DT+GNN**
>
> > What is the loss of information when constructing a categorical state space of DT+GNN using O(n) bits to encode the information from the continuous embeddings?
>
> In theory we are looking at an exponential loss of information when switching from O(n) categorical to O(n), as we need O(n) categorical states/bits to encode a number that otherwise could be represented with O(log(n)) bits. Fortunately, Table 1a) shows that this has little impact in practice: important information in the datasets is compressible enough such that the problems can be solved with a much smaller domain.
>
> > Is there a hard threshold on the number of decision leaves per tree, which is restricted in the training process?
>
> Yes, we train trees that have at most 100 leaves in the DT+GNN distillation process.
>
> > Most DT+GNN architecture use more than two layers for synthetic and real-world datasets. It is unclear whether both GIN and DT+GNN use these many layers during training. If yes, then it is very counterintuitive that GIN needs 5 layers to get high performance for small datasets like BA-Shapes and Tree-Cycles.
>
> We added a more expansive discussion of the number of layers to section 4.2 and added it. GIN trains with 5 layers and embeddings of size 16. DT+GNN also trains with 5 layers and a categorical state space of 10. After training we took a look at the resulting trees to see how many layers and states were actually used. We retrain DT+GNN with those numbers and these numbers are in Table 1b).
>
> It is well possible that GIN does not make use of all 5 layers in all cases (especially where we can see from the principally weaker DT+GNN that fewer layers are enough). In this case, GIN can also make use of the skip connections we have to the output layer to skip non-important layers. Due to this adaptivity we did not perform an ablation on the number of layers for GIN. As this should in principle be determined during learning.
>
> > The limited number of categorical states in DT+GNN limits its applicability to large benchmark graph datasets.
>
> We see three possible perspectives that define the largeness of graph dataset: 1) the number of graphs in the dataset; 2) the (mean) number of nodes in each individual graph; 3) the number of input features. If the first two definitions are concerned DT+GNN we see no hindrance for scaling DT+GNN to larger datasets. For example, we can see that DT+GNN works reasonably well for decently sized benchmarks such as REDDIT-BINARY or COLLAB which have roughly 80000 and 350000 nodes respectively. What DT+GNN indeed struggles with is scaling to large feature input dimensions. We added more experiments in Appendix E: Generally, Diff-DT+GNN produces decent results even when the number of features or the graph size becomes large. However, distilling to DT+GNN is where the accuracy drops clearly on all datasets. Intuitively this makes sense as we aim to have shallow trees and fewer states in the model to keep it interpretable. Shallow trees are also inherently limited in the number of features they can process (a tree with 100 leaves can only have 99 splits on features). We experimented with more leaves per tree (up to 500) on OBGN-Arxiv, which improves accuracy but trees are no longer interpretable.

---

### Author Response · Authors · 2022-11-13
**Overview of changes to the Rebuttal Version**

We want to thank the reviewers for your time and reviews. Based on your feedback we made some changes to the paper and want to highlight the updates here:

* We expanded the introduction on the difference of importance score versus DT+GNN. We added a new example on the PROTEINS dataset to highlight the differences.
* We rearranged and merged chapters 3.1 and 3.2. The new chapter now also contains mathematical formulations of all layers.
* The paper includes a new appendix for discussion and experiments of high-dimensionality and large datasets (Cora, CiteSeer, PubMed, OGBN-Arxiv).

---

### Decision · Program_Chairs · 2023-01-20

**Decision:**

Reject

**Justification For Why Not Higher Score:**

It is not very clear how the proposed method can lead to good prediction performance as well as understandable decision process for practical problems.

**Justification For Why Not Lower Score:**

N/A

**Metareview: Summary, Strengths And Weaknesses:**

This work address the interpretability issue of GNNs. First it uses the gumble-softmax function to induce categorical latent states for GNN, the second part uses model distillation to replace the key functions in the GNN with decision trees. The authors claim that the resulting distilled model will not only provide importance scores but also allow users to inspect the decision process.
Strength:
- The use of discrete states, coupled with decision tree distillation of the GNN functions offers the potential for human to inspect the decision process
- The resulting model is more interpretable compared to the continuous counterpart.
- Results show that the importance score it generates align well with human intuition
- Some qualitative evidence of interpretable decision process on small data sets

weakness:
- distillation leads to performance loss on large datasets
- interpretation results limited to smaller datasets
- the main claim of providing interpretable decision process is questionable --- given the latent states and many layers of message passing (hence many DTs with different inputs of latent states) --- reviewers are not convinced that the proposed method will allow the user to understand the decision process, at least not easily. The example provided by the authors in their response are all for small datasets where one already knows what to looking for more or less.



**Summary Of Ac-Reviewer Meeting:**

Discussed the claim that the work provides interpretable decision process and found it not well supported.